# Antibiotic-Induced Gut Microbial Dysbiosis Reduces the Growth of Weaning Rats via FXR-Mediated Hepatic IGF-2 Inhibition

**DOI:** 10.3390/nu16111644

**Published:** 2024-05-27

**Authors:** Yan Wang, Shuai Ma, Mindie Zhao, Lei Wu, Ruqian Zhao

**Affiliations:** Key Laboratory of Animal Physiology and Biochemistry, College of Veterinary Medicine, Nanjing Agricultural University, Nanjing 210095, China; wy1311156742@163.com (Y.W.); 2019207039@stu.njau.edu.cn (S.M.); zmd9839@163.com (M.Z.); leiwu@njau.edu.cn (L.W.)

**Keywords:** FXR, IGF-2, gut microbiota, antibiotic, rat

## Abstract

The gut microbiota plays a crucial role in postnatal growth, particularly in modulating the development of animals during their growth phase. In this study, we investigated the effects of antibiotic-induced dysbiosis of the gut microbiota on the growth of weaning rats by administering a non-absorbable antibiotic cocktail (ABX) in water for 4 weeks. ABX treatment significantly reduced body weight and feed intake in rats. Concurrently, ABX treatment decreased microbial abundance and diversity in rat ceca, predominantly suppressing microbes associated with bile salt hydrolase (BSH) activity. Furthermore, decreased appetite may be attributed to elevated levels of glucagon-like peptide-1 (GLP-1) in the serum, along with reduced neuropeptide Y (NPY) and increased cocaine and amphetamine-regulated transcript (CART) in the hypothalamus at the mRNA level. Importantly, concentrations of insulin-like growth factor 1 (IGF-1) and insulin-like growth factor 2 (IGF-2) were decreased in the serum and liver of antibiotic-treated rats. These alterations were associated with significant down-regulation of IGF-2 mRNA in the liver and significantly decreased farnesoid X receptor (FXR) protein expression and binding to the IGF-2 promoter. These results indicate that antibiotic-induced gut microbial dysbiosis not only impacts bile acid metabolism but also diminishes rat growth through the FXR-mediated IGF-2 pathway.

## 1. Introduction

Gut microbiota, in a symbiotic relationship with its host, plays a crucial role in the host’s growth and development [1]. The remodeling of gut microbiota during early growth stages significantly influences the development of the animal. It has been observed that newborn intrauterine growth restriction (IUGR) piglets exhibit a lower diversity of gut microbiota [2]. In addition, research has shown that the body weight gain of eight-week-old germ-free mice was lower than that of conventional mice [3], and mixed antibiotic-treated six-week-old mice had lower body weights and body fat in a high-fat diet model [4]. These studies demonstrate the crucial role of gut microbes in regulating growth performance. While the metabolic effects of gut microbiota on adult animals due to antibiotic treatment have been widely studied, its impact on the development and growth of the critical window period of animals remains less explored.

Insulin-like growth factor (IGF) is a pivotal growth factor controlling postnatal growth in mammals [1]. Studies have shown that the circulating level of IGF-1 is significantly higher in mice with an intact gut microbiota compared to those in germ-free mice [5]. The study found that the mice gut microbiota impacted bone formation via the generation of short-chain fatty acids (SCFAs), which then promoted host IGF-1 production [6]. Additionally, individual probiotics also have the effect of promoting growth. Lactobacillus plantarum strains have been demonstrated to promote growth by activating the growth hormone/insulin-like growth factor 1 (GH/IGF-1) axis in chronically malnourished mice [5]. Furthermore, the gut microbiota modulates important appetite-suppressing hormones like GLP-1 through metabolites, primarily short-chain fatty acids [7] and secondary bile acids (secondary BAs) [8], affecting the diet by altering nutrient perception and signaling from the gut to the brain.

Primary bile acids (primary BAs) are metabolized by the gut bacteria to increase the diversity and lipophilicity through various enzymatic modifications including deconjugation, dehydroxylation, desulfation, oxidation/epimerization, esterification, and amidation [9]. All the major gut microbiota phyla, including Bacteroidetes, Firmicutes, and Actinobacteria, possess bile salt hydrolase activity that deconjugates primary bile acids to provide the bacteria with the energy from the amino acid and to decrease the BA toxicity for the host [10]. Deconjugated BAs are further dehydroxylated and epimerized by 7α-dehydroxylases, oxidases, and epimerases to form secondary BAs [9]. Gut microbiota regulates bile acid metabolism in an FXR-dependent manner, with gut microbiota inhibiting hepatic bile acid synthesis in mice by reducing bile acid tauro-β-muricholic acid (T-βMCA) production and activating the farnesoid X receptor–fibroblast growth factor 15 (FXR-FGF15) pathway [11]. Bile acids are crucial metabolites of the gut microbiota and act as signaling molecules to regulate body growth and feeding, binding to glucose-dependent insulinotropic receptor 119 (GPR119) and G-protein-coupled bile acid receptor (TGR5) in the gastrointestinal tract to regulate the secretion of appetite-related hormones such as glucagon-like peptide-1 (GLP-1) and peptide YY (PYY) [8,12]. Additionally, dietary bile acid supplementation increases breast muscle growth by increasing FXR protein expression and binding to the IGF-2 promoter in broilers [13]. Nevertheless, the specific impact of gut microbiota on growth through the modulation of the bile acid nuclear receptor FXR remains to be fully elucidated.

Therefore, this study aimed to investigate the effects of antibiotic-induced gut microbial dysbiosis on the growth performance of rats and to explore the possible mechanisms by determining the serum hormones and hepatic expression of genes involved in growth regulations and bile acid metabolism, as well as the status of these changes in relation to FXR-mediated regulation of IGF-2 in rats. This study provides insights into the underlying targets for the regulation of growth and development in animals during critical windows of growth.

## 2. Materials and Methods

### 2.1. Ethics Statement

The experimental protocol was approved by the Animal Ethics Committee of Nanjing Agricultural University. The project number is NJAU. No. 20221025199. The sampling procedures followed the “Guidelines on Ethical Treatment of Experimental Animals” (2006) No. 398 set by the Ministry of Science and Technology, China.

### 2.2. Animals and Experimental Design

Twenty male Wistar rats (21 days old) weighing 46 g on average were purchased from Beijing Vital River Laboratory Animal Technology Co., Ltd., Beijing, China. Rats were housed under controlled temperature (22 °C  ±  0.5 °C) and humidity (50  ±  5%) with artificial lighting (12L:12D light–dark cycle). Subsequently, rats were randomly divided into control (CON) and antibiotic (ABX) groups. The CON group was fed the basal diet and provided normal drinking water, while the ABX group was fed the same diet and drinking water with a non-absorbable antibiotic cocktail (1 g/L ampicillin, 1 g/L neomycin, 1 g/L metronidazole, and 500 mg/L vancomycin) for 4 weeks. Antibiotics were shielded from light and refreshed daily. All rats had free access to water and diet throughout the experiment. After 4 weeks of treatment, all rats were fasted overnight before euthanasia. The organ weights were weighed by an accurate electronic scale, while the organ index was calculated using the following equation: organ index (%) = organ weight/body weight × 100. Blood samples were collected and centrifugated at 3000 rpm for 10 min to separate the serum stored at −20 °C for further analysis. The samples including liver, ileum, hypothalamic, and cecum contents were rapidly frozen in liquid nitrogen and stored at −80 °C for subsequent analyses.

### 2.3. Determination of Total Bile Acids and Cholesterol Concentration

The concentrations of total bile acid in the liver and cecum contents were determined using a previously established method [14]. Simply, 50 mg of liver or cecum contents were homogenized with 75% ethanol, incubated in a 50 °C water bath for 2 h, and then centrifuged at 8000 rpm for 10 min. The chloroform–methanol extraction method was used to measure total cholesterol levels in liver tissues. Briefly, liver and cecum contents were homogenized, and then the cholesterol was extracted by homogenizing it with chloroform–methanol (2:1 *v*/*v*). The extracts and serum were used for detecting the total bile acids (TBA, H101T) and cholesterol (CHOL, H202) concentrations with an automatic biochemical analyzer (Hitachi 7020, Tokyo, Japan) by using respective commercial kits purchased from Meinkang Biotech Co., Ltd. (Ningbo, China). The intra- and inter-assay coefficients of variation for total bile acids were 5%. The intra- and inter-assay coefficients of variations of the cholesterol kit were 3% and 5%, respectively.

### 2.4. Measuring Concentrations of IGFs

Briefly, 20 mg of liver tissue was homogenized with 180 μL of phosphate-buffered saline (PBS, pH = 7.4) and then centrifuged at 3000 rpm for 20 min. The supernatant was collected and stored in a −80 °C freezer for later use. IGF-1 (YB-IGF1-RT) and IGF-2 (YB-IGF2-RT) concentrations in the liver homogenates and serum were determined with the respective enzyme-linked immunosorbent assay (ELISA) kits purchased from Shanghai Yubo Biotechnology (Shanghai, China). The intra- and inter-assay coefficients of variations of all the kits were 9% and 11%, respectively.

### 2.5. 16S rDNA Gene Amplicon Sequencing

Bacterial genomic DNA was extracted using a fecal genome extraction kit (D4015-01, Omega Bio-Tek, Norcross, GA, USA). PCR amplification of the V3–V4 region of the bacterial 16S rDNA gene was performed using primer 341F (5′-CCTAYGGRBGCASCAG-3′) and primer 806R (5′-GGACTACNNGGGTATCTAAT-3′). PCR products were purified using the AxyPrep DNA gel extraction kit (NA1111-1KT, Axygen Biosciences, Union City, CA, USA) and quantified using Quantus™ Fluorometer (Promega, Beijing, China). Subsequently, high-throughput pyrosequencing of PCR products was performed on a NovaSeq PE250 platform at Novogene Co., Ltd. (Beijing, China).

All high-quality reads from one sample were clustered into the operational taxonomic unit (OTU) with a ≥97% similarity cutoff using UPARSE (version 7.12) software. For alpha diversity analysis, the OTU was rarified into several metrics, including curves of OTU rank, rarefaction, and Shannon, and the indexes of Shannon, Chao1, and Abundance-based Coverage Estimator (ACE) were calculated. For beta diversity analysis, QIIME1 was used to analyze the principal coordinates of the key OTUs identified by RDA.

### 2.6. Bile Acid Analysis

The targeted bile acid metabolomics were used for measuring the major primary and secondary bile acids and their respective glycine and taurine conjugates. One-hundred-milligram cecum contents were homogenized in 400 μL of methanol and then centrifuged at 12,000 rpm for 10 min at 4 °C. The supernatant was collected and then filtered by a 0.22 μm nylon syringe filter. The mobile phase flow rate was 0.25 mL/min, and the injection volume was 5 μL. Different components of bile acids were separated by gradient elution. A negative selective ion monitoring mode was selected to acquire all mass spectrometry data, which were analyzed by the Xcalibur 4.0 software. The above bile acid analysis was carried out by Shanghai Personal Biotechnology (Shanghai, China).

### 2.7. RNA Extraction and Real-Time PCR

Thirty milligrams of the liver, hypothalamus, and ileal samples was extracted using 1 mL of Trizol (Invitrogen, Carlsbad, CA, USA) to obtain high-quality RNA, and 1 μg of total RNA was reverse transcribed to cDNA using random hexamer primers (RK20429, ABclonal Technology Co., Wuhan, China). Two microliters of diluted cDNA (1:20) were amplified with real-time PCR (RK21206, ABclonal Technology Co., Wuhan, China). Peptidylprolyl isomerase A (PPIA), which was not affected by the treatment, was chosen as a reference gene. Data were analyzed by using the method of 2^−ΔΔCT^ [15]. All primers (Appendix A) were synthesized by Tsingke Biotechnology (Beijing, China).

### 2.8. Western Blotting

Frozen liver and ileum tissues were homogenized in a RIPA lysis buffer with 1% protease inhibitor cocktails (P1010, Beyotime, Shanghai, China), then quantified using the BCA protein concentration determination kit (DQ111-01, TransGen, Nanjing, China). The protein samples were analyzed and quantitated after sodium dodecyl sulfate-polyacrylamide gel electrophoresis (SDS-PAGE). The primary antibody FXR was used for target protein determination (A8320, ABclonal, China, dilution 1:1000). Tubulin α was used as the internal control (BS1699, Bioworld, Nanjing, China, dilution 1:10,000). Coomassie Brilliant Blue was used to stain whole protein bands as an internal reference for ileum tissue. Briefly, the protein gel was stained overnight in approximately 100 mL of Coomassie Brilliant Blue R250 staining solution, followed by incubation on a shaker with destaining solution until the bands became apparent.

### 2.9. Chromatin Immunoprecipitation Assay

Two hundred milligrams of frozen liver tissue was washed with PBS containing a protease inhibitor cocktail (P1010, Beyotime, Shanghai, China). After crosslinking in 1% formaldehyde, the reaction was stopped with glycine. After the tissue was fully lysed, the chromatin was sonicated to an average length of approximately 250 bp. Chromatin was diluted in ChIP dilution buffer and incubated with FXR antibodies (sc-25309X, Santa Cruz, CA, USA) overnight. Protein G agarose beads (sc-2003, Santa Cruz, CA, USA) were added to capture immunoprecipitated chromatin complexes, and purified DNA fragments were used as real-time PCR templates. The sequences of putative FXRE are amplified using the specific primers listed in Appendix A.

### 2.10. Statistical Analysis

Data are expressed as means ± SEM. Differences between the two groups were analyzed with a *t*-test using SPSS 26.0 software (SPSS Inc., Chicago, IL, USA). The results for average daily water intake, average daily feed intake, and body weight were analyzed using a general linear model. The differences were considered statistically significant when *p* < 0.05.

## 3. Results

### 3.1. Growth Performance

Average daily water intake was not affected by ABX treatment (Figure 1A). Notably, for the average daily feed intake (Figure 1B), the effects of antibiotics (*p* < 0.001), time (*p* < 0.001), and the interaction between antibiotics and time (*p* = 0.008) are all significant. Similar results were observed for body weight (Figure 1C), with the antibiotic effect (*p* < 0.001), and time effect (*p* < 0.001), and the interaction between antibiotics and time (*p* = 0.08) all being significant. Moreover, the caecum index in the ABX treatment increased by 98.28% (*p* < 0.05), and the epididymal fat index decreased by 39.93% (*p* < 0.05) (Figure 1D).

### 3.2. Microbiota Composition in Cecum Contents

The principal component analysis (PCA) results (Figure 2A) showed a significant difference in microbiota structure in cecum contents between the control and ABX groups. ABX treatment significantly decreased the means of Chao1 (Figure 2B) and ACE (Figure 2C) (*p* < 0.05). Similarly, the Shannon diversity index (Figure 2D) was significantly decreased in the ABX group compared to the CON group (*p* < 0.05). Linear discriminant analysis effect size (LEfSe) analysis (Figure 2E) exhibited that the relative abundance of different levels of Enterobacter was significantly higher in the cecum contents of the ABX group (*p* < 0.05). The common function of some microbial genera is the production of BSH enzymes which are used to deconjugate glycine- or taurine-conjugated bile acids to form unconjugated bile acids. The reduction in BSH-producing bacteria (Figure 2F) was evident in ABX treatment (*p* < 0.05).

### 3.3. Biochemical Parameter Concentrations and Bile Acid Metabolomics

Compared with the CON group, the cholesterol concentrations in serum (Figure 3A), liver (Figure 3B), and cecum contents were increased by 10.31%, 10.28%, and 36.04% (*p* < 0.05) (Figure 3C). Concurrently, total bile acids concentrations in the serum (Figure 3D), liver (Figure 3E), and cecum contents (Figure 3F) were decreased by 55.01%, 41.26%, and 91.39% (*p* < 0.05), which may be associated with significantly increased cholesterol contents (*p* < 0.05). The principal component analysis (PCA) (Figure 3G) revealed a distinct clustering of bile acids for each experimental group. Remarkable changes in the bile acid composition of cecum contents were induced by ABX treatment. Heatmap analysis (Figure 3H) exhibited that antibiotic treatment significantly decreased the levels of most bile acids in cecum contents (*p* < 0.01). In contrast, taurocholic acid (TCA), tauro-α-muricholic acid (T-αMCA), and tauro-β-muricholic acid (T-βMCA) were significantly increased in the ABX group (*p* < 0.05) (Figure 3I).

### 3.4. Expression of Feeding-Related Genes

ABX treatment decreased the protein expression of FXR (Figure 4A) in the ileum (*p* < 0.01). In addition, glucagon (*Gcg*) mRNA expression was up-regulated (Figure 4B) and dipeptidyl peptidase 4 (*DPP-4*) mRNA expression (Figure 4C) was down-regulated in the ileum of ABX treatment group. (*p* < 0.05). Serum glucagon-like peptide-1 (GLP-1) concentration (Figure 4D) in the ABX treatment group was increased by 18.39% (*p* < 0.05). Moreover, the mRNA expression of glucagon-like peptide-1 receptor (*GLP-1R*) (Figure 4E) in the hypothalamus, a receptor in the hypothalamus that binds to GLP-1, also increased significantly (*p* < 0.05). Concurrently, the mRNA expression of *NPY* in the hypothalamus significantly decreased (*p* < 0.05), while the mRNA expression of *CART* was higher in ABX treatment rats (*p* < 0.05) (Figure 4F).

### 3.5. Expression of Growth-Related Genes and FXR Binding Levels in the Promoter Region of IGF-2

Serum IGF-1 (*p* = 0.080) (Figure 5A) and IGF-2 (*p* = 0.078) (Figure 5B) concentrations in ABX treatment were decreased by 21.24% and 10.69%, while hepatic IGF-1 (Figure 5C) and IGF-2 (Figure 5D) concentrations were decreased by 33.26% and 30.28% (*p* < 0.05). The mRNA expression of hepatic IGF-2 (Figure 5E) in the ABX treatment group was significantly down-regulated (*p* < 0.05). The protein expression of hepatic FXR (Figure 5F) tended to decrease (*p* = 0.06). In addition, ChIP-PCR analysis revealed a significant decrease in FXR binding to the fragment of the IGF-2 gene promoter region (*p* < 0.05) (Figure 5G).

## 4. Discussion

The gut microbiota is currently considered a key regulator of host energy metabolism [16]. Gut microbiota modulates the body’s immunity in pathological states, while it is equally essential for growth and development under normal physiological conditions [5,17]. The weight of germ-free mice is less than conventional mice with normal microbiota [5]. In addition, germ-free (GF) mice colonized with the gut microbiota of conventional (CONV) mice had increased body fat and decreased insulin sensitivity [18]. In this study, mixed antibiotic treatments significantly reduced body weight in rats. Antibiotic treatment decreased food intake in rats, which is not consistent with previous studies in germ-free mice. This may be due to the different models that disrupt the gut microbiota, related to the dose and time of antibiotic use. Interestingly, antibiotic treatment increased cecum weight in rats, which may be due to disturbed cecum microbiota, leading to incomplete cecum fermentation and water retention, which is caused by the accumulation of unfermented contents [19,20].

Antibiotic treatment as an inexpensive and readily available method of microbiome regulation aims to mimic the reduction in microbial diversity to induce homeostatic dysregulation [21]. Our study verified that antibiotic treatment significantly reduced the abundance and diversity of the bacterial flora in the cecum contents. However, antibiotic treatment did not completely deplete the rat gut microbiota, and Enterobacter under the phylum Proteobacteria was significantly increased at different levels. Enterobacter is recognized as a major drug-resistant bacterial pathogen and opportunistic pathogen of plants, animals, and humans [22]. Antibiotic treatment decreased the abundance of the phylum Firmicutes and Bacteroidota and increased the abundance of Proteobacteria in the feces of mice, which is consistent with our results [23]. However, it is unclear whether Enterobacter is involved in regulating the growth of antibiotic-treated rats. Gut microbiota regulates bile acid metabolism through the uncoupling of bile acids by bacteria with bile salt hydrolase activity [24]. Our results indicate that antibiotic treatment reduces the abundance of Bacteroidetes, Lactobacillus, and Ligilactobacillus with functional BSH activity.

In this study, antibiotic treatment increased cholesterol content and decreased bile acid concentration in the serum, liver, and cecum contents of rats. It appears that antibiotic treatment decreased the diversity and content of the bile acid pool, which may be due to gut microbiota having profound effects on bile acid metabolism by promoting deconjugation, dehydrogenation, and dehydroxylation of primary bile acids in the distal small intestine [24]. Our results are consistent with reduced levels of serum and fecal bile acids in germ-free mice. However, the elevated levels of the total bile acid pool in germ-free mice may be due to elevated bile acids in the gallbladder [11]. Rats do not have a gallbladder, and bile was not collected from the bile ducts in this study; therefore, the changes in the total bile acid pool are unknown. In addition, unlike germ-free mice, acute and dramatic depletion of the microbiome caused by antibiotic treatment may also affect the regulation of metabolic homeostasis [23].

In this study, antibiotic treatment decreased the content of most bile acids and significantly increased the content of T-α/β MCA and TCA in the contents of rat cecum. Interestingly, T-βMCA was significantly increased in the blood, liver, gallbladder, small intestine, cecum contents, and feces of germ-free mice [11]. It appears that the abundance of bacteria with bile salt hydrolase activity is reduced, decreasing the deconjugation of conjugated bile acids by the gut microbiota [25]. Bile acids affect host metabolism through signaling by FXR and TGR5, of which T-βMCA is an antagonist of FXR [26]. In this study, antibiotic treatment reduced protein expression of FXR in rat liver and ileum. As mentioned above, antibiotic-induced gut microbial dysbiosis regulates enterohepatic circulation and hepatic bile acid/cholesterol metabolism by up-regulating T-βMCA levels and decreasing FXR expression in the ileum. High levels of serum GLP-1 are a potent appetite suppressant, slowing intestinal transportation and affecting satiety [27,28]. In this experiment, we found that antibiotic treatment up-regulated the mRNA expression of glucagon (Gcg) in the ileum, increased serum GLP-1 levels in rats, bound to GLP-1 receptors in the hypothalamus, and inhibited the mRNA expression of the appetite-promoting gene NPY. It is noted that the intestinal bile acid nuclear receptor FXR regulates GLP-1. FXR up-regulates miR-33 expression in STC-1 cells. miR-33 binding to Gcg 3′UTR degrades Gcg mRNA, thereby inhibiting the secretion of GLP-1 [29]. Therefore, it is likely that antibiotic treatment inhibits appetite in rats by FXR-mediated GLP-1 regulation.

The intestinal microbiota is known to influence postnatal growth [1]. Microbiota has recently been shown to induce host IGF-1 synthesis to influence growth [5]. Metabolites produced by microorganisms such as short-chain fatty acids may induce IGF-1 production [6]. However, there are few studies on the regulation of IGF-2 by gut microbiota and its mechanisms. In our study, antibiotic treatment reduced IGF-1 and IGF-2 levels in the serum and liver of rats; this was accompanied by a significant reduction in FXR binding to the fragment of the IGF-2 gene promoter region. Similar to our previous findings, dietary bile acid supplementation improves breast muscle growth in broilers fed a high-fat diet, through the FXR-mediated IGF-2 pathway [13]. Therefore, we speculate that the growth-regulating IGF-2 gene is regulated by the bile acid nuclear receptor FXR in the liver.

This study provides the phenotypic changes of rats in response to 4 weeks of antibiotic treatment, linking gut microbiota and bile acid composition with IGF-2 production and secretion. Based on the results, we hypothesize that disrupted gut microbiota and BA homeostasis in the ABX group induce FXR-mediated IGF-2 inhibition, thereby causing reduced body weight. However, the exact relationship between these events needs further research. Future studies may be directed at screening microbial regulators to mitigate the adverse effects of antibiotics on the gut flora and to improve animal growth performance. Moreover, as GLP-1 is considered a therapeutic target for obesity and diabetics in human clinics, the mechanism underlying increased serum GLP-1 concentration in ABX-treated rats warrants further investigation.

## 5. Conclusions

In conclusion, our findings reveal that antibiotic treatment inhibits rat growth. Specifically, antibiotics inhibit growth possibly through the FXR/IGF2 pathway. This study demonstrates the gut–liver responses to the abuse of antibiotics and provides possible targets, including FXR/IGF2, for screening probiotics and other microbiome regulators to alleviate gut microbiota disturbance caused by antibiotics, or promoting the animal growth inhibited by transient antibiotic treatment in veterinary practice.

## Figures and Tables

**Figure 1 nutrients-16-01644-f001:**
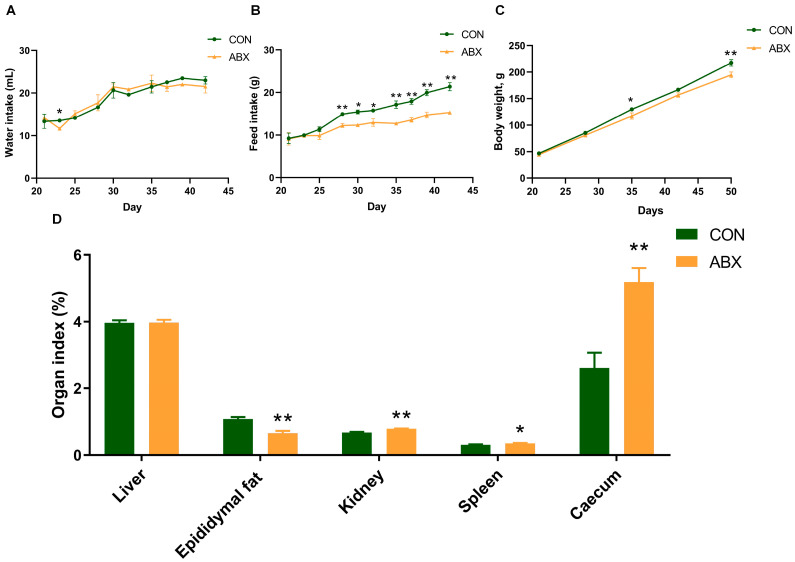
Growth performance. (**A**) Water consumption line chart at 21 to 42 days of age, a cage as the statistical unit, *n* = 2, 5 rats per cage; (**B**) feed intake line chart at 21 to 42 days of age, a cage as the statistical unit, *n* = 2, 5 rats per cage; (**C**) body weight line chart, the number of rats as the statistical unit, *n* = 10; (**D**) organ index = organ weight/body weight × 100. Values are mean ± SEM, * *p* < 0.05, ** *p* < 0.01, control (CON) groups, and antibiotic cocktail (ABX) groups (*n* = 10).

**Figure 2 nutrients-16-01644-f002:**
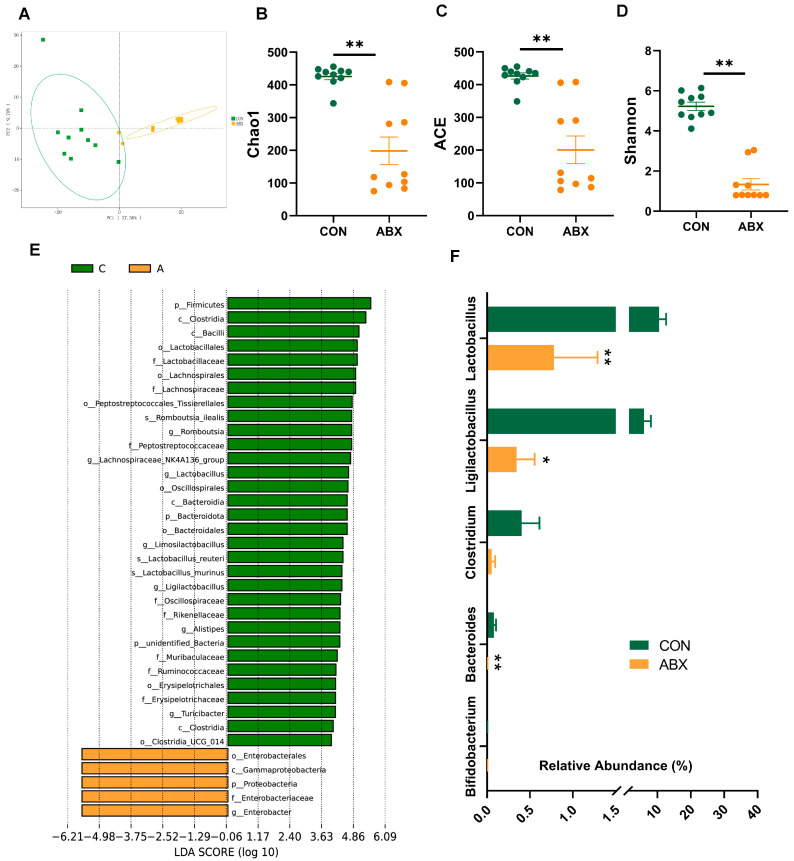
Microbiota composition in cecum contents. (**A**) The principal component analysis (PCA) of the gut microbiota; (**B**) Chao1 of the gut microbiota, green dots are CON and orange dots are ABX; (**C**) Abundance-based Coverage Estimator (ACE) of the gut microbiota; (**D**) Shannon’s diversity index (Shannon) of the gut microbiota; (**E**) linear discriminant analysis effect size (LEfSe) analysis; (**F**) abundance of bile salt hydrolase (BSH)-producing bacteria. Values are mean ± SEM, * *p* < 0.05, ** *p* < 0.01, control (CON) groups, and antibiotic cocktail (ABX) groups (n = 10).

**Figure 3 nutrients-16-01644-f003:**
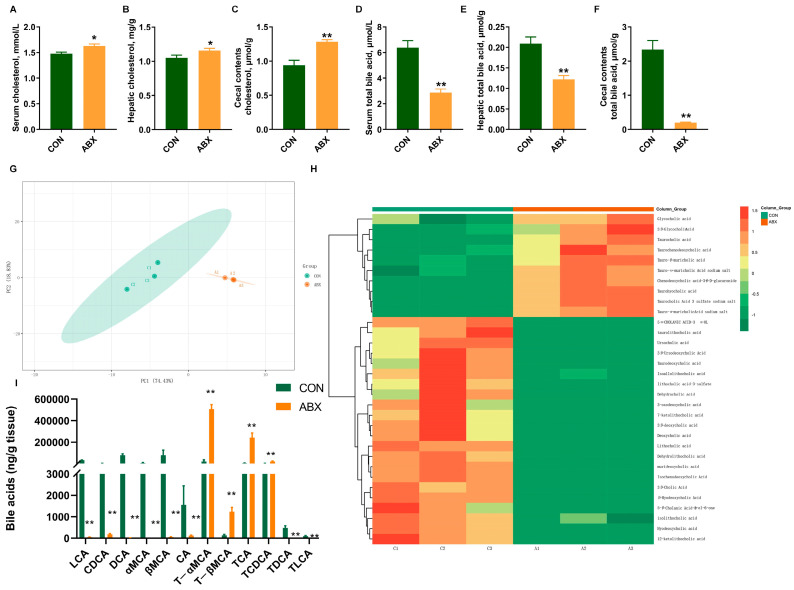
Biochemical parameter concentrations and bile acid metabolomics. (**A**) Serum cholesterol concentration, *n* = 10; (**B**) hepatic cholesterol concentration, *n* = 10; (**C**) cecum contents cholesterol concentration, *n* = 10; (**D**) serum total bile acids concentration, *n* = 10; (**E**) hepatic total bile acids concentration, *n* = 10; (**F**) cecum contents total bile acids concentration, *n* = 10; (**G**) principal component analysis of bile acid metabolomics, *n* = 3; (**H**) heatmap analysis of bile acid metabolomics, *n* = 3; (**I**) the contents of specific bile acids in cecum contents, *n* = 3. Values are mean ± SEM, * *p* < 0.05, ** *p* < 0.01, control (CON) groups, and antibiotic cocktail (ABX) groups.

**Figure 4 nutrients-16-01644-f004:**
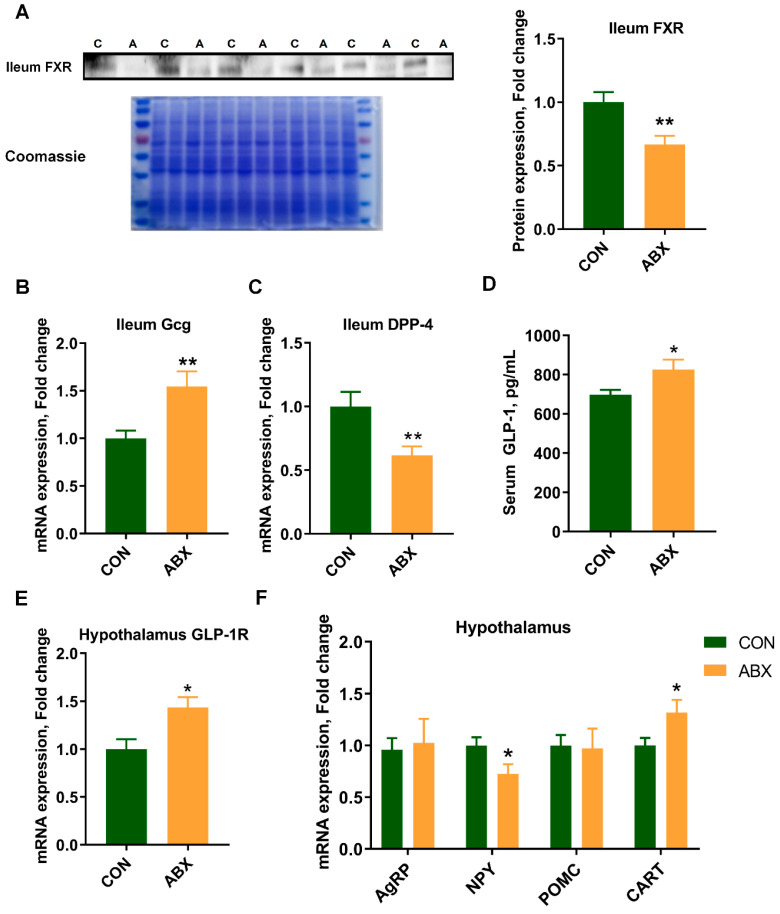
Expression of feeding-related genes. (**A**) Ileum farnesoid X receptor (FXR) protein expression, Coomassie stain full strips as internal reference, *n* = 6, CON (C) group and ABX (A) group; (**B**) ileum glucagon (Gcg) gene mRNA expression, *n* = 10; (**C**) ileum dipeptidyl peptidase 4 (DPP-4) gene mRNA expression, *n* = 10; (**D**) serum glucagon-like peptide-1 (GLP-1) concentration, *n* = 10; (**E**) hypothalamus glucagon-like peptide-1 receptor (GLP-1R) gene mRNA expression, *n* = 10; (**F**) hypothalamus feeding-related gene neuropeptide Y (NPY), cocaine and amphetamine-regulated transcript (CART), agouti-related protein (AgRP) and proopiomelanocortin (POMC) mRNA expression, *n* = 10. Values are mean ± SEM, * *p* < 0.05, ** *p* < 0.01, control (CON) groups, and antibiotic cocktail (ABX) groups.

**Figure 5 nutrients-16-01644-f005:**
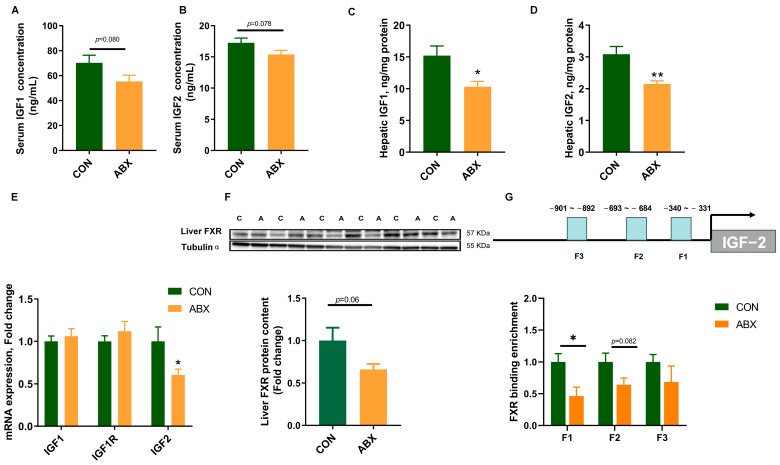
Expression of growth-related genes and FXR binding levels in promoter region of IGF-2. (**A**) Serum insulin-like growth factor 1 (IGF-1) concentration, *n* = 10; (**B**) serum insulin-like growth factor 2 (IGF-2) concentration, n = 10; (**C**) hepatic IGF-1 concentration, *n* = 10; (**D**) hepatic IGF-2 concentration, *n* = 10; (**E**) liver growth-related gene IGF-1, IGF-2 and insulin-like growth factor 1 receptor (IGF-1R) mRNA expression, *n* = 10; (**F**) liver FXR protein expression, *n* = 6; (**G**) schematic representation of FXR transcription factor binding promoter of IGF-2 and ChIP-PCR assay was used to measure the binding of FXR on IGF-2 promoter in the liver, *n* = 4. Values are mean ± SEM, * *p* < 0.05, ** *p* < 0.01, control (CON) groups, and antibiotic cocktail (ABX) groups.

## Data Availability

Data are contained within the article.

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
