# Peer review of "Antibiotic-Induced Gut Microbial Dysbiosis Reduces the Growth of Weaning Rats via FXR-Mediated Hepatic IGF-2 Inhibition"

_nutrients, 2024, doi:10.3390/nu16111644_

Round 1

Reviewer 1 Report

Comments and Suggestions for Authors

The manuscript by Wang Y. shows the effect of non-absorbable antibiotic cocktail (ABX) on several parameters in weaning rats. The findings cannot suggest any mechanism by which the gut microbial dysbiosis affect growth. The authors should change the conclusions of their study.

The authors should spelled out all the abbreviation throughout the study.

The measurement of organ index is mainly derived from the lower weight of the rats.

Figure 1 should be better presented. The average daily water intake should follow the line chart. Similarly, the average feed intake should follow the feed intake line chart. Then the Body weight.

The index should be explained in the figure legend. 

Figure 2:  the legend should explain what is con and abx and all the other abbreviations.

The authors assumed that the lower feed intake was related to the change in gut microbiome and they based their assumption on another study (ref 19), but it is still possible that the taste of the antibiotics had an effect of the food intake.      

Author Response

Dear reviewer,

We are grateful to the reviewers for all the comments and valuable suggestions. We have followed carefully all the comments and made changes to the manuscript according to the suggestions.

Here are the details of the response to the reviewers’ comments, and the amendments are highlighted in yellow in the revised manuscript. Please see the attachment.

Sincerely yours,

Professor Ruqian Zhao

Reviewer 2 Report

Comments and Suggestions for Authors

The manuscript, entitled “Antibiotic-induced gut microbial dysbiosis reduces the growth  2 of weaning rats via FXR-mediated hepatic IGF-2 inhibition” is well written.” Nevertheless, The manuscript could be improved.

The manuscript should provide be more details on the “Materials and Methods.” Much of this section can be expanded. 

Many of the figures, the authors mention should explain the origin of error bars.

Percentage of the data, average, standard deviation?

The “Results and Discussion” section is not well articulated.

Specific comments

Introduction

Line 49-50: Authors should expand on the role of gut microbiota on bile acid metabolism.

Bacteria, enzymes what conditions are important?

Materials and methods

Line 78: Can you please justify why Wistar rats were used.

Any references on the methodology of animal model?

Line 107-108: please provide details on the procedure. What manufacturer's instructions.

Please clarify…

Line 144-145: please provide a reference on how the data was normalize with PPIA.

Line 170-172: In saditical analysis,

Why no general linear model was applied to study the concentration intake effect, time effect and the interaction or regression analysis to have a depth analysis on the data?

Results and discussion

Overall, Please improved the discussion.

What are the implication and possible hypothesis of the results?

What are the recommendations for future studies?

Conclusion

Overall, Please improved the conclusion.

Why the finding of the study important.

What are recommendations to improve the understanding of microbial dysbiosis on reducing the growth  of weaning rats via FXR-mediated hepatic IGF-2 inhibition?

Author Response

(The authors gave the same response as above.)

Round 2

Reviewer 2 Report

Comments and Suggestions for Authors

Corrections were made. 

Accept manuscript in present form.